# Study on a Hexagonal Acoustic Metamaterial Cell of Multiple Parallel-Connection Resonators with Tunable Perforating Rate

**DOI:** 10.3390/ma16155378

**Published:** 2023-07-31

**Authors:** Hongxiang Cheng, Fei Yang, Xinmin Shen, Xiaocui Yang, Xiaonan Zhang, Shaohua Bi

**Affiliations:** 1Air Force Second Aviation Equipment Training Base, Shenyang 110000, China; 2Field Engineering College, Army Engineering University of PLA, Nanjing 210007, China; 19962061916@163.com (F.Y.); zxn826@163.com (X.Z.); 17337434454@163.com (S.B.); 3Engineering Training Center, Nanjing Vocational University of Industry Technology, Nanjing 210023, China; 2019101052@niit.edu.cn; 4MIIT Key Laboratory of Multifunctional Lightweight Materials and Structures (MLMS), Nanjing University of Aeronautics and Astronautics, Nanjing 210016, China

**Keywords:** hexagonal acoustic metamaterial cell, tunable perforating rate, adjustable sound absorption performance, joint optimization, acoustic finite element simulation, experimental validation

## Abstract

The limited occupied space and various noise spectrum requires an adjustable sound absorber with a smart structure and tunable sound absorption performance. The hexagonal acoustic metamaterial cell of the multiple parallel-connection resonators with tunable perforating rate was proposed in this research, which consisted of six triangular cavities and six trapezium cavities, and the perforation rate of each cavity was adjustable by moving the sliding block along the slideway. The optimal geometric parameters were obtained by the joint optimization of the acoustic finite element simulation and cuckoo search algorithm, and the average sound absorption coefficients in the target frequency ranges of 650–1150 Hz, 700–1200 Hz and 700–1000 Hz were up to 0.8565, 0.8615 and 0.8807, respectively. The experimental sample was fabricated by the fused filament fabrication method, and its sound absorption coefficients were further detected by impedance tube detector. The consistency between simulation data and experimental data proved the accuracy of the acoustic finite element simulation model and the effectiveness of the joint optimization method. The tunable sound absorption performance, outstanding low-frequency noise reduction property, extensible outline structure and efficient space utilization were favorable to promote its practical applications in noise reduction.

## 1. Introduction

Annoying noise has become one of the causes for social instability and led to substantial societal burden beyond the health impacts [1,2,3,4,5,6]. It was reported by Hener [1] that the increasing background noise by 4.1 decibels caused a 6.6% increase in the violent crime rate and the additional crimes mostly consisted of physical assaults on humans. Karabey [2] had investigated the relationship between noise sensitivity and violent tendencies among nursing students, which pointed out that a raise of 0.203 units on the violence tendencies scale might be expected per unit increase on the noise sensitivity scale. The effects of chronic exposure to noise on aggression in automotive industry workers was studied by Alimohammadi et al. [3], who concluded that the exposure to noise in the work environment increased the incidence of tension and inappropriate behavior associated with aggression. Shiue [4] had pointed out that the people who lived in towns or cities tended to indicate certain major problems, such as noise, air quality, low quality of drinking water, crime and/or violence, rubbish and traffic congestion. The research achievements gained by Mesene et al. [5] indicated that noise pollution affected social relations and quality of living environment, disturbed spoken communication and work performance, had a short and long effect on hearing capacity, disturbed mental health, decreased satisfaction with life, etc. Li et al. [6] had investigated the spatial distributions of multiple types of noise pollution perceived by residents in Beijing, and the results showed that perceived higher noise pollution exposure was significantly associated with worse mental health, which supplied effective implications on the policy implications for environmental pollution mitigation and healthy city development in China. The prevention and control of noise pollution is beneficial to the sustainable development of the economy and the construction of a harmonious society, which attracts research attention in both natural sciences and social sciences.

One of the major difficulties in noise pollution control is that its frequency spectrum is not completely fixed, which requires the spectrum of noise reduction of a sound absorber to be broadband [7,8,9,10,11,12,13,14,15,16]. It had been reported by Gao et al. [7] that the ultra-broadband parallel sound absorber showed high sound absorption from 200 to 1715 Hz by optimizing the lengths of both lateral and micro-perforated plates. Since conventional sound absorbers could hardly have good performance of low-frequency and broadband absorption simultaneously, the gradually perforated porous materials backed with a Helmholtz resonant cavity were proposed by Liu et al. [8]. An acoustic metasurface was constructed by Long et al. [9] by integrating three types of coiled space resonators and coupling an ultrathin sponge coating, and the deep-subwavelength broadband absorber with high absorptivity (>80%) exceeding one octave from 185 Hz to 385 Hz was demonstrated experimentally. An ultra-thin meta-absorber was proposed by Zhang and Cheng [10] to achieve broadband low-frequency underwater sound absorption via inserting thin and thickness-graded circular-elastic–plate scatterers (CPSs) into an elastomer matrix. Cheng et al. [11] designed a composite loaded sound absorber composed of a sub-wavelength Helmholtz resonator and porous material, and a broadband quasi-perfect sound absorption of more than 0.9 between 160 Hz and 285 Hz was realized by optimization. Gao and Zhang [12] had applied a teaching–learning-based optimization algorithm to obtain the average sound absorption coefficient 0.938 in 0 to 1600 Hz with the optimized size of the lateral plates, micro-perforated plates, and cavities. A broadband low-frequency absorber based on an acoustic metaporous composite was developed by Xu et al. [13], and the high sound absorption coefficient exceeding 0.8 was achieved within a broad frequency range of 290–1074 Hz. The single-layer and double-layer micro-perforated panel backed by the cavity were optimized by Li et al. [14], which obtained an absorption level of at least 0.8 in the frequency range of 794–1614 Hz and 632–1954 Hz, respectively. Wu et al. [15] had studied the impedance matching of composite acoustic metamaterials comprising micro-perforated plates and subsequent Fabry–Perot channels, and an average sound absorption coefficient greater than 0.9 in the frequency range of 250–4500 Hz was obtained with the total thickness of 100 mm. A multiple layer microperforated panel absorber was optimized by Bucciarelli et al. [16], and it was demonstrated that the 5-layer microperforated panel absorber could guarantee a high absorption (constantly over 90%) in the frequency range from 400 to 2000 Hz. These proposed and developed sound absorbers [7,8,9,10,11,12,13,14,15,16] improve the acoustic environment effectively and stably promote social development.

Metamaterial has been developed to obtain the unique performance by the clever structural design and precision manufacturing relative to original natural materials [17,18,19,20]. In order to reduce the occupied space of a sound absorber with the desired sound absorption property, acoustic metamaterials with online tunable parameters have been developed [21,22,23,24,25,26], which can satisfy the requirements of limited total thickness and various noise spectrum, simultaneously. Adjustable parallel Helmholtz acoustic metamaterial had been developed by Yang et al. [21], and the target for all sound absorption coefficients above 0.9 was achieved in 602−1287 Hz with normal incidence, and the target for all sound absorption coefficients above 0.85 was obtained in 618−1482 Hz. Xing et al. [22] had designed the adjustable membrane-type acoustic metamaterials with negative pressure cavity, and the advantages of this structure were that the location of the sound absorption peak could be adjusted. The origami-based foldable sound absorber based on micro-perforated resonators was developed by Jiang et al. [23]; the effective absorption of which could be realized via a design whose average thickness was only 1/34.4 lambda for the resonance frequency. An adjustable sound absorber of multiple parallel-connection Helmholtz resonators with tunable apertures was proposed by Yang et al. [24]; the actual average sound absorption coefficients of which with a total thickness of 40 mm for the frequency ranges 500–800 Hz, 550–900 Hz, 600–1000 Hz and 700–1150 Hz reached 0.9203, 0.9202, 0.9436 and 0.9561, respectively. Xu et al. [25] had proposed a tunable low-frequency acoustic absorber composed of multi-layered ring-shaped microslit tubes with a deep subwavelength thickness, which had the advantages of tunable functionality, compactness, high efficiency, wide-angle absorption and easy fabrication. The open hollow sphere model with a negative equivalent elastic modulus and hollow tube model with a negative equivalent mass density had been coupled into a whole by Zhai et al. [26], the relative impedance of which could be changed by simply rotating the inner splitting ring around the axis, and the position of the absorption peak could be freely controlled in a wide frequency band.

Therefore, a hexagonal acoustic metamaterial cell of the multiple parallel-connection resonators with tunable perforating rate was proposed and investigated in this research, which aimed to develop a practical sound absorber for noise reduction. Firstly, the 3-dimensional model of a hexagonal acoustic metamaterial cell was proposed and designed, which exhibited its overall structure and the working principle. Secondly, the theoretical model and acoustic finite element simulation model of the hexagonal acoustic metamaterial cell were constructed, which could conduct the initiatory study and preliminary verification of the sound absorption performance. Thirdly, taking the actual noise reduction requirement into account, the geometric parameters of the hexagonal acoustic metamaterial cell were optimized, which could satisfy the different requirement through adjusting the perforating rate of each cavity. Fourthly, the experimental sample was fabricated, and its actual sound absorption coefficients were detected, which validated the effectiveness of the proposed acoustic metamaterial and the accuracy of the joint optimization method. Meanwhile, the feasibility and practicability of this tunable acoustic metamaterial cell was further investigated, which provided effective guidance for its practical application in the noise reduction field with the varying noise spectrum. The major advantages of this hexagonal acoustic metamaterial cell were the adjustable sound absorption performance, flexible tunable structure, minor occupied space waste, excellent extensibility and expandability, outstanding low-frequency noise reduction property, etc.

## 2. Materials and Methods

The 3-dimensional model of a hexagonal acoustic metamaterial cell of multiple parallel-connection resonators with the tunable perforating rate was constructed first, which supplied the intuitive understanding of its working principle. Afterwards, the theoretical model of the proposed acoustic metamaterial was built based on the Helmholtz resonance principle, which could preliminarily verify the feasibility. Later, the actual noise reduction requirements were taken into account, and the corresponding optimal parameters were obtained by the joint optimization combined of the finite element method and cuckoo search algorithm. Finally, the sample was fabricated by the low force stereolithography method, and its actual sound absorption coefficients were detected according to the transfer function method, which could validate the effectiveness of these optimized parameters.

### 2.1. Structural Design

The 3-dimensional model of the hexagonal acoustic metamaterial cell is shown in Figure 1, which mainly consists of the basic chamber and 12 sliding blocks. It could be observed from Figure 1a that there were 6 triangular cavities and 6 trapezium cavities in the basic chamber, which were labelled C1 to C6 and T1 to T6, respectively, as shown in Figure 1b. Taking the triangular cavity C3 for example, its 3-dimensional model and corresponding sliding block are shown in Figure 1c and Figure 1d, respectively. Meanwhile, for the trapezium cavity T4, its 3-dimensional model and the corresponding sliding block are shown in Figure 1e and Figure 1f, respectively. The sliding block could move back and forth along the slideway in the cavity, which could adjust the cross-sectional area of the quadrangle opening zone, which was the corresponding aperture for each resonator. To gain excellent low-frequency sound absorption performance by increasing the lengths of the apertures, the lateral plates were added to the boundary areas, as shown in Figure 1a,c,e. For the triangular cavity shown in Figure 1c, the aperture was formed by the 2 lateral plates, the sliding block in Figure 1d and the dividing board. Regarding the trapezium cavity shown in Figure 1e, the aperture was formed by the lateral plate, the dividing board, the side plate and the sliding block in Figure 1f. Meanwhile, the size of each triangular cavity was equal, and that of each trapezium cavity was the same as well, which was favorable to reduce the optimization difficulty of geometric parameters and improve the structural stability of the experimental sample.

The size of the hexagonal acoustic metamaterial cell is shown in Figure 2. Taking the requirement of actual noise reduction circumstance and that of the standing wave tube detection into consideration, the length of the side of the hexagonal acoustic metamaterial cell and the width of each wall in the chamber were set as 50 mm and 2 mm, respectively, so the total size of the sample could be limited to the Φ 100 mm, as shown in Figure 2a, which was the diameter of the tube in the standing wave detection. As mentioned above, the space was equally divided into 6 triangular cavities and 6 trapezium cavities. Meanwhile, for the 6 triangular cavities, the width and the thickness of the aperture were set as 5.5 mm and 10 mm, respectively, as shown in Figure 2b,c, and the length of the aperture could be adjusted with the optional value in the range of 6–11.17 mm by moving the sliding block. Similarly, for the 6 trapezium cavities, the width and the thickness of the aperture were set as 10.44 mm and 12 mm, respectively, as shown in Figure 2d,e, and the length of the aperture could be adjusted with the optional value in the range of 6.94–14.14 mm through moving the sliding block. For both the triangular cavity and trapezium cavity, the area of each cavity was calculated as 423.725 mm^2^, which would be equivalent to a cylindrical cavity with the diameter of 23.23 mm.

### 2.2. Theoretical Model

The theoretical sound absorption coefficient *α* of the hexagonal acoustic metamaterial cell could be calculated by Equation (1), which was based on the Helmholtz resonance principle [27,28,29] according to the electro-acoustic theory [30,31,32]. In Equation (1), the symbols of $Z_{am}$, $\rho_{0}$ and $c_{0}$ were the total acoustic impedance of the hexagonal acoustic metamaterial cell, the density of the air (1.21 kg/m^3^) and the acoustic velocity in air (343 m/s), respectively. As shown in Figure 1, the hexagonal acoustic metamaterial cell was composed of 6 triangular cavities and 6 trapezium cavities with the same sectional area, which could be considered the multiple parallel-connection of 12 single Helmholtz resonators, so the $Z_{am}$ could be derived by Equation (2). Here $Z_{n}$ was the acoustic impedance of each single Helmholtz resonator, and it consisted of the acoustic impedance of the front aperture $Z_{na}$ and that of the rear cavity $Z_{nc}$, as shown in Equation (3).
(1)$$\alpha =1-{\left|\frac{Z_{am}-\rho_{0}c_{0}}{Z_{am}+\rho_{0}c_{0}}\right|}^{2}$$
(2)$$Z_{am}=1/\sum_{n=1}^{12}\left(\frac{1}{Z_{n}}\right)$$
(3)$$Z_{n}=Z_{na}+Z_{nc}$$

For the acoustic impedance $Z_{na}$ of the front aperture, it could be calculated by Equation (4) according to the Euler equation [33,34]. In Equation (4), the symbols of ω, $l_{n}$, $\sigma_{n}$, $B_{1}\left(\eta_{n}\sqrt{-i}\right)$, $B_{0}\left(\eta_{n}\sqrt{-i}\right)$, $\eta_{n}$, μ and $d_{n}$ were the sound angular frequency (which could be derived by Equation (5)), the length of the aperture (the values were 10 mm for the triangular cavity and 12 mm for the trapezium cavity), the perforation rate (which could be derived by Equation (6)), the first order Bessel functions of the first kind, the zero order Bessel functions of the first kind, the perforation constant (which could be derived by Equation (7)), the dynamic viscosity coefficient of the air (1.8 × 10^−5^ Pa·s) and the equivalent diameter of the front aperture through equating the rectangle aperture to the round aperture (which could be derived by Equation (8)), respectively.
(4)$$Z_{na}=i\frac{\omega \rho_{0}l_{n}}{\sigma_{n}}{\left[1-\frac{2B_{1}\left(\eta_{n}\sqrt{-i}\right)}{\left(\eta_{n}\sqrt{-i}\right)\cdot B_{0}\left(\eta_{n}\sqrt{-i}\right)}\right]}^{-1}+\frac{\sqrt{2}\mu \eta_{n}}{\sigma_{n}\cdot d_{n}}+i\frac{0.85\omega \rho_{0}\cdot d_{n}}{\sigma_{n}}$$
(5)ω=2πf
(6)$$\sigma_{n}=\frac{L_{n}W_{n}}{\frac{3\sqrt{3}}{2}a_{0}^{2}}$$
(7)$$\eta_{n}=d_{n}\sqrt{\frac{\rho_{0}\omega }{4\mu }}$$
(8)$$d_{n}=\sqrt{\frac{4L_{n}W_{n}}{\pi }}$$

In Equation (5), f was the frequency of the incident acoustic wave. In Equation (6), $L_{n}$ was the length of the rectangle aperture, which could be adjusted by moving the sliding block, and its ranges were 6–11.17 mm for the triangular cavity and 6.94–14.14 mm for the trapezium cavity; $W_{n}$ was the width of the rectangle aperture, and the values were 5.5 mm for the triangular cavity and 10.44 mm for the trapezium cavity; $a_{0}$ was the side length of the hexagonal acoustic metamaterial cell, and its value was 50 mm.

Meanwhile, the acoustic impedance $Z_{nc}$ of the rear cavity could be derived by Equation (9) according to the impedance transfer formula [35]. In Equation (9), the effective density $\rho_{0e}$ and the effective volumetric compressibility $C_{0e}$ of the air could be derived by Equations (10) and (11), respectively, and T was the thickness of the cavity. Moreover, in Equations (10) and (11), v could be derived by Equation (12); A was the sectional area of each cavity, and its value was 423.725 mm^2^ in this research; the intermediate calculation coefficients $\alpha_{x}$ and $\beta_{y}$ could be derived by Equations (13) and (14), respectively; $P_{0}$ was the standard atmospheric pressure under normal temperature and its value was 1.01325 × 10^5^ Pa; γ was the specific heat rate of the air and its value was 1.4; and $v^{\prime }$ could be derived through Equation (15). In Equation (15), the κ and $C_{v}$ were the thermal conductivity and the specific heat capacity, respectively, and their corresponding values were 0.0258 W/(m·K) and 718 J/(kg·K).
(9)$$Z_{nc}=-i\sqrt{\frac{\rho_{0e}}{C_{0e}}}\cot\left(\omega \sqrt{\rho_{0e}C_{0e}}T\right)$$
(10)$$\rho_{0e}=\rho_{0}\frac{vA^{2}}{4i\omega }{\left\{\sum_{x=0}^{\infty }\sum_{y=0}^{\infty }{\left[\alpha_{x}^{2}\beta_{y}^{2}\left(\alpha_{x}^{2}+\beta_{y}^{2}+\frac{i\omega }{v}\right)\right]}^{-1}\right\}}^{-1}$$
(11)$$C_{0e}=\frac{1}{P_{0}}\left\{1-\frac{4i\omega \left(\gamma -1\right)}{v^{\prime }A^{2}}{\sum_{x=0}^{\infty }\sum_{y=0}^{\infty }\left[\alpha_{x}^{2}\beta_{y}^{2}\left(\alpha_{x}^{2}+\beta_{y}^{2}+\frac{i\omega \gamma }{v^{\prime }}\right)\right]}^{-1}\right\}$$
(12)$$v=\frac{\mu }{\rho_{0}}$$
(13)$$\alpha_{x}=\frac{\left(x+1/2\right)\pi }{\sqrt{A}}$$
(14)$$\beta_{y}=\frac{\left(y+1/2\right)\pi }{\sqrt{A}}$$
(15)$$v^{\prime }=\frac{\kappa }{\rho_{0}C_{v}}$$

Based on these Equations (1)–(15), the theoretical sound absorption coefficients of the proposed hexagonal acoustic metamaterial cell could be obtained, which could exhibit its sound absorption performance in the low-frequency range. The symbols utilized in this study and their corresponding meanings are summarized in Table A1 in Appendix A. However, it could be observed that there were some approximations, equivalences, simplifications, and omissions in the theoretical modeling process, which indicated that the prediction accuracy of the theoretical model might be limited. Fortunately, the acoustic finite element simulation could simulate the actual propagation process of the sound wave more realistically, which could improve the prediction accuracy effectively [36,37,38] and was utilized to predict and analyze the sound absorption coefficients of the proposed acoustic metamaterial.

### 2.3. Finite Element Simulation

The acoustic finite element simulation model used to investigate the sound absorption performance of the proposed hexagonal acoustic metamaterial cell was first constructed in the commercial COMSOL Multiphysics 5.5 software, as shown in Figure 3, which consisted of a perfect matching layer, background acoustic field and hexagonal acoustic metamaterial cell. After the mesh partition, the initial model in Figure 3a was transferred to the gridded model in Figure 3b. In order to improve the prediction accuracy, the extremely fine mesh was selected, and the maximum unit size, the minimum unit size, the maximal unit growth rate, the curvature factor and the resolution of the narrow region were 2 mm, 0.02 mm, 1.3, 0.2 and 1, respectively. The corresponding labels of the 6 triangular cavities and 6 trapezium cavities are shown in Figure 3c, the orders and the relative positions of which were consistent with those in Figure 1b. Meanwhile, the gridded model of the hexagonal acoustic metamaterial cell is shown in Figure 3d. Based on the built finite element simulation model, the sound absorption performance of the proposed acoustic metamaterial could be predicted, and the influences of the parameter changes could be investigated as well.

Moreover, the sound absorption property of each triangular cavity and that of each trapezium cavity were investigated, which could provide effective guidance for parametric selection to obtain the expected sound absorption performance of the whole hexagonal acoustic metamaterial cell. The selected length of the aperture was in the range of 6–11 mm with an interval of 1 mm for the triangular cavity and was in the range of 7–14 mm with an interval of 1 mm for the trapezium cavity. At the same time, the other parameters of the finite element simulation models in Figure 4 were completely consistent with those in Figure 3.

The sound absorption performances for the single cavity are shown in Figure 5. It was found that along with the increase in the length of the aperture, the resonance frequencies of both the triangular cavity and the trapezium cavity shifted towards the high frequency range, which is consistent with the normal sound absorption properties of a single Helmholtz resonator [39,40,41].

The resonance frequencies and corresponding peak sound absorption coefficients for the single cavity are summarized in Table 1. It could be observed that the resonance frequencies for the single triangular cavity ranged from 662 to 874 Hz, gradually, with the length of the aperture *L* increasing from 6 to 11 mm and that the single trapezium cavity ranged from 906 to 1187 Hz with the length of the aperture *L* increasing from 6 to 11 mm, which indicated that their noise reduction spectrum ranges could be linked by selecting the appropriate geometric parameters and the possible sound absorption frequency range could reach 650–1200 Hz. Meanwhile, it could be observed that the peak sound absorption coefficients were different for the various resonance frequencies because the resonance noise reduction effect of the Helmholtz resonator would reach its maximum impact with the appropriate match of its geometric parameters [39,40,41]. Thus, for the expected sound absorption performance, the desired collocation of the perforation rates for the hexagonal acoustic metamaterial cell could be gained by adjusting the length of the aperture, *L,* for each cavity in the reasonable ranges.

### 2.4. Parameter Optimization

There were 2 kinds of parameters for the proposed hexagonal acoustic metamaterial cell. First were the selected parameters, such as the length of side of the hexagonal acoustic metamaterial cell, the width of each wall in the chamber, the width and thickness of the aperture for the 6 triangular cavities and those for the 6 trapezium cavities, etc. These parameters were selected according to the actual requirements of the noise reduction and the experimental measurements. Second was the adjustable parameter, which consisted of the lengths of rectangle apertures for the 12 cavities. In this study, the optimization of the adjustable parameter was conducted by the joint optimization of the acoustic finite element simulation and the cuckoo search algorithm instead of the experimental design methods in the selection of variables and their levels. The normal experimental design method, such as the orthogonal experimental method, was suitable for the condition that the influencing laws of parameters were not obvious. In this study, the influencing laws of parameters to the sound absorption coefficients were on a regular basis, so the initial values of parameters could be determined by the desired sound absorption properties, and their optimal parameters could be derived by iterative optimization in the cuckoo search algorithm. Therefore, the joint optimization of the acoustic finite element simulation and the cuckoo search algorithm was applied to obtain the optimal parameters of length of the apertures, which could promote the optimization efficiency and accuracy.

For each cavity, the perforation ratio was determined by the length of the aperture *L*, and it was symbolled as *L_n_* (*n* = 1, 2, …, 6) for the 6 triangular cavities and *L_n_* (*n* = 7, 8, …, 12) for the 6 trapezium cavities. The initial structure of the hexagonal acoustic metamaterial cell with the step perforation rates was investigated first, which selected the length of the aperture with the equal interval for the triangular cavity and for the trapezium cavity, respectively, as shown in Table 2. The length of the aperture for the triangular cavity was in the range of 6–11 mm with an interval of 1 mm and for the trapezium cavity was in the range of 7–14.1 mm with an interval of 1.4 mm. It can be judged from Figure 5 and Table 1 that the effective sound absorption performance of the hexagonal acoustic metamaterial cell was in the range of 600–1250 Hz, which took the minimum resonance frequency of 662 Hz and the maximum resonance frequency of 1187 Hz into account. Thus, the sound absorption property of the hexagonal acoustic metamaterial cell could be tuned in this range by adjusting the length of each aperture for the 12 cavities. A total of 3 target sound absorption frequency ranges were taken as examples, which were 650–1150 Hz, 700–1200 Hz and 700–1000 Hz. A joint optimization of the acoustic finite element simulation and cuckoo search algorithm was applied to obtain the optimal geometric parameters of the length of the apertures [37], and the average sound absorption coefficient for the target frequency range was taken as the criterion to evaluate the performance for each group of the geometric parameters. The acoustic finite element simulation was based on the constructed model in Figure 3 and the geometric parameters were updated by the cuckoo search algorithm [42,43,44], which could promote the optimization efficiency and improve the optimization accuracy. The joint optimization was stopped when the changes in average sound absorption coefficient for the target frequency range in the past 10 optimization processes were smaller than 0.01 or the number of the iterative optimization reached 10,000. The obtained appropriate geometric parameters of the hexagonal acoustic metamaterial cell for the 3 target frequency ranges are summarized in Table 2.

The theoretical sound absorption coefficients of the hexagonal acoustic metamaterial cell for certain specific scenarios are shown in Figure 6. It could be observed that there were 10 resonance frequencies for the initial structure of the hexagonal acoustic metamaterial cell with the step perforation rates, which were 663 Hz, 709 Hz, 755 Hz, 800 Hz, 842 Hz, 882 Hz, 960 Hz, 1025 Hz, 1093 Hz and 1169 Hz. Moreover, when the target frequency range was 650–1150 Hz, there were 11 resonance frequencies, which were 664 Hz, 689 Hz, 737 Hz, 791 Hz, 836 Hz, 882 Hz, 920 Hz, 962 Hz, 1011 Hz, 1080 Hz and 1140 Hz; when the target frequency range was set as 700–1120 Hz, there were 9 resonance frequencies, which were 722 Hz, 763 Hz, 807 Hz, 850 Hz, 884 Hz, 968 Hz, 1026 Hz, 1104 Hz and 1170 Hz; and when the target frequency range was 700–1000 Hz, there were 8 resonance frequencies, which were 723 Hz, 750 Hz, 790 Hz, 847 Hz, 896 Hz, 919 Hz, 948 Hz and 979 Hz. Some of the resonance frequencies generated by the cavities were coupled into 1 sound absorption peak, which was consistent with the normal sound absorption principle of the parallel connection of multiple Helmholtz resonators [45,46,47]. Furthermore, the average sound absorption coefficients of the hexagonal acoustic metamaterial cell in the target frequency ranges of 650–1150 Hz, 700–1200 Hz and 700–1000 Hz were 0.8565, 0.8615 and 0.8807, respectively; all of which were larger than the average sound absorption coefficient of the initial structure of the hexagonal acoustic metamaterial cell with the step perforation rate of 0.7261 in the frequency range of 600–1250 Hz. With the same noise reduction bandwidth of 500 Hz, the average sound absorption coefficient for the target frequency range of 650–1150 Hz was smaller than for the target frequency range of 700–1200 Hz, which was consistent with the normal sound absorption principle that it is easier to obtain better sound absorption performance in the high frequency range. Meanwhile, it was easier to achieve a larger average sound absorption coefficient for the smaller noise reduction bandwidth with the same lower limit of the frequency range, which could be judged from the sound absorption performance for the target frequency range of 700–1200 Hz and for the target frequency range of 700–1000 Hz. Thus, it could be judged from the sound absorption performance of the hexagonal acoustic metamaterial cell for the certain specific scenarios that it was tunable for the target frequency range by adjusting the perforation rates of each cavity.

### 2.5. Sample Fabrication

According to the selected geometric parameters of the hexagonal acoustic metamaterial cell in Figure 2, the experimental sample for the hexagonal acoustic metamaterial cell was fabricated by the fused filament fabrication method [48,49], which was realized by the Raise 3D Pro2 Plus printer (Shanghai Fusion Tech Co., Ltd., Shanghai, China), as shown in Figure 7. The hexagonal acoustic metamaterial cell was divided into 4 parts, which included the chamber, the rear panel, the 6 sliding blocks for the triangular cavity and 6 sliding blocks for the trapezium cavity, and they were prepared by the 3D printer one by one. Afterwards, the experimental sample was assembled, as shown in Figure 7b. In order to distinguish between different structures, the chamber and rear panel were fabricated by red wire, and the 12 sliding blocks were fabricated by yellow wire. Meanwhile, the sealing silicone oil was smeared at the interfaces between the chamber and the sliding blocks, which aimed to avoid the sound leakage. By this method, the perforation rate of each cavity could be adjusted by moving the sliding blocks in the chamber.

### 2.6. Sound Absorption Coefficient Detection

The experimental sample of the hexagonal acoustic metamaterial cell was further detected by the AWA6290T impedance tube detector (Hangzhou Aihua Instruments Co., Ltd., Hangzhou, China) to obtain the actual sound absorption coefficients [50,51,52], as shown in Figure 8. The hexagonal acoustic metamaterial cell was installed at the end of the impedance tube, and it was supported by the sample holder so that there was no air gap in this measuring process. The incident sound wave and reflected sound wave were detected by the 2 microphones, and the signals were transmitted to the workstation through the dynamic signal analyzer to derive the actual sound absorption coefficients. The distance between the detected hexagonal acoustic metamaterial cell and microphone 2 was kept at 100 mm, and the distance between the 2 microphones was selected as 70 mm, which realized the investigated frequency range of 200–1600 Hz according to the standard of the GB/T 18696.2–2002 (ISO 10534–2:1998) ‘Acoustics–Determination of sound absorption coefficient and impedance in impedance tubes–part 2: Transfer function method’ [53]. By this method, the actual sound absorption coefficients at the 1502 sampling frequency points in the frequency range of 200–1600 Hz could be obtained. There were, in total, 200 times of measurement at each frequency point, and the final data were an average of the 200 values, which was conducive to eliminate the accidental error and improve the detection accuracy. For each specific scenario, the corresponding perforation rates were adjusted according to the parameters in Table 2, and the sample was detected singly.

## 3. Results and Discussions

### 3.1. Comparative Analysis

For the four certain specific scenarios, the comparative analysis of the sound absorption performance in the simulation with that of the actual system was conducted, as shown in Figure 9. It could be observed that the basic trend of the simulation data was consistent with that of the experimental data for each specific scenario, which proved the accuracy of the acoustic finite element simulation model and the effectiveness of the joint optimization method. Meanwhile, it could be seen that the actual resonance frequencies moved towards the high frequency range. There were three major reasons for this phenomenon. Firstly, the aperture for each cavity was not a standard cuboid in the actual system, which could be judged from the two kinds of sliding blocks in Figure 1 and Figure 2. Thus, the actual volume of the aperture in the experimental sample was larger than that in the simulation model in Figure 3, which led to the shift of the resonance frequency towards the high frequency range. Secondly, fabrication error was inevitable in the 3D printing process, especially when the dimensions of the part were small, which indicated a larger relative error. The fabrication error would result in the misregistration of the geometric parameters for the 12 cavities in the hexagonal acoustic metamaterial cell, which caused the resonance frequency shift towards the high frequency range as well. Thirdly, although the acoustic finite element simulation tried to imitate the actual impedance tube detection process as much as possible, there were differences between the two. In the acoustic finite element simulation model, the face of each surface was smooth, but the actual face of the experimental sample was rough. The deviation between simulation data and experimental data was reasonable and explicable.

### 3.2. Sound Absorption Mechanism

Taking the optimized structure for the target frequency ranges of 650–1150 Hz as an example, the sound absorption mechanism of the hexagonal acoustic metamaterial cell was revealed by the distributions of the sound pressure for the various resonance frequency [54,55,56,57], as shown in Figure 10. The 11 resonance frequencies were 664 Hz, 689 Hz, 737 Hz, 791 Hz, 836 Hz, 882 Hz, 920 Hz, 962 Hz, 1011 Hz, 1080 Hz and 1140 Hz. Comparing with the model of the hexagonal acoustic metamaterial cell in Figure 3c, it could be seen that these 11 resonance frequencies were chiefly generated by the C2, C3, C4, C5, C6, T1, T1 and T2, T2 and T3, T4 and T5, T5 and T6, and T6 cavities, successively. The cavity C1 assisted in generating the first resonance frequency at 664 Hz, since the generation of sound absorption peak at the low-frequency range was difficult to realize. Thus, it could be found that all 12 cavities contributed to the outstanding integral sound absorption performance of the whole hexagonal acoustic metamaterial cell, and the final sound absorption effect resulted from the coupling actions of the 12 individual Helmholtz resonators. Moreover, it could be found that the generation of each resonance frequency was relatively independent and could be realized by one cavity with the assistance of its neighboring cavities. Thus, the adjustment of each resonance frequency could be realized by mainly changing the perforation rates of the corresponding cavity and fine tuning the neighboring cavities, which was accomplished by moving the sliding blocks in the cavities. It should be noted that the neighboring cavities were the cavities with similar perforation rates instead of those adjacent in space. The revealed sound absorption mechanism further proved the feasibility of the hexagonal acoustic metamaterial cell.

## 4. Conclusions

The hexagonal acoustic metamaterial cell of multiple parallel-connection resonators with tunable perforating rates was proposed, and the major achievements in structural design, parametric analysis, sample realization and mechanism study were as follows:(1)The hexagonal acoustic metamaterial cell consisting of six triangular cavities and six trapezium cavities was proposed, and the perforation rate of each cavity was adjustable by moving the sliding block along the slideway in the cavity. The acoustic metamaterial with tunable sound absorption performance was developed by this method, and the optional ranges of length of the aperture were 6–11.17 mm for the triangular cavity and 6.94–14.14 mm for the trapezium cavity.(2)The optimal geometric parameters of length of the apertures were obtained by the joint optimization of the acoustic finite element simulation and cuckoo search algorithm, and the average sound absorption coefficients in the target frequency ranges of 650–1150 Hz, 700–1200 Hz and 700–1000 Hz were 0.8565, 0.8615 and 0.8807, respectively, which exhibited an excellent tunable sound absorption performance.(3)The experimental sample for the hexagonal acoustic metamaterial cell was fabricated by the fused filament fabrication method, which included the chamber, rear panel, six sliding blocks for the triangular cavity and six sliding blocks for the trapezium cavity. Its sound absorption coefficients were further detected by the impedance tube detector. The actual average sound absorption coefficients for the target frequency ranges of 650–1150 Hz, 700–1200 Hz and 700–1000 Hz reached 0.8391, 0.8443 and 0.8759, respectively.(4)The consistency between the simulation data and the experimental data proved the accuracy of the acoustic finite element simulation model and the effectiveness of the joint optimization method. The revealed sound absorption mechanism by the distributions of the sound pressure at the resonance frequencies of 664 Hz, 689 Hz, 737 Hz, 791 Hz, 836 Hz, 882 Hz, 920 Hz, 962 Hz, 1011 Hz, 1080 Hz and 1140 Hz further proved the feasibility of the hexagonal acoustic metamaterial cell.

The developed hexagonal acoustic metamaterial cell had the advantages of adjustable sound absorption performance, excellent low-frequency noise reduction property, extensible outline structure, and efficient space utilization, which was conducive to the promotion of its practical applications in the field of sound absorption for the varying noise spectrum.

## Figures and Tables

**Figure 1 materials-16-05378-f001:**
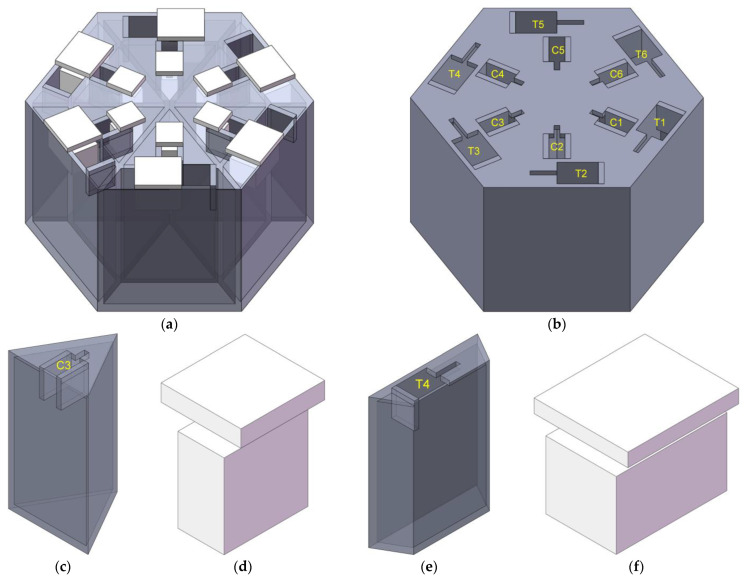
The 3-dimensional model of the hexagonal acoustic metamaterial cell. (**a**) The whole structure; (**b**) the basic chamber; (**c**) the triangular cavity; (**d**) the sliding block for the triangular cavity; (**e**) the trapezium cavity; and (**f**) the sliding block for the trapezium cavity.

**Figure 2 materials-16-05378-f002:**
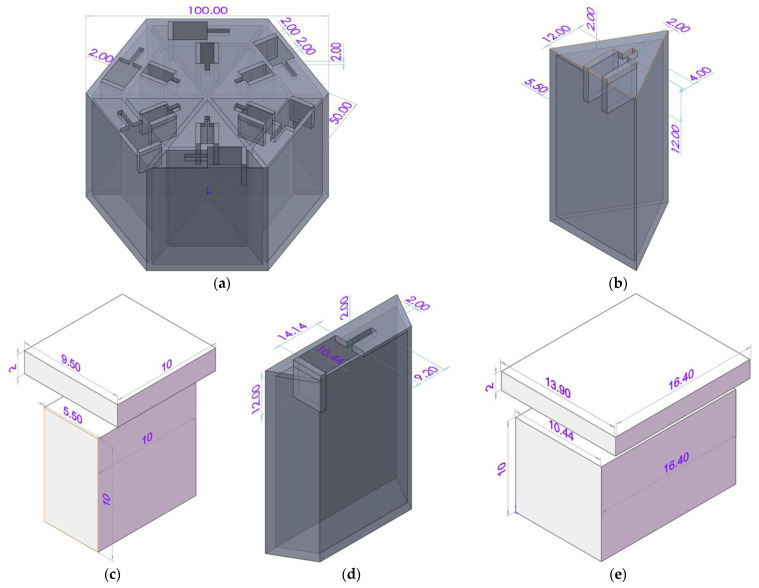
The size of the hexagonal acoustic metamaterial cell with the unit of mm for all the marked values. (**a**) The chamber; (**b**) triangular cavity; (**c**) sliding block for triangular cavity; (**d**) trapezium cavity; and (**e**) sliding block for trapezium cavity.

**Figure 3 materials-16-05378-f003:**
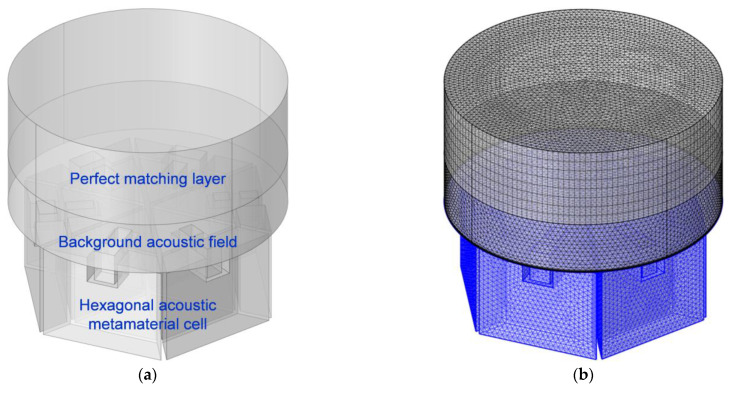
The acoustic finite element simulation model to investigate the sound absorption performance of the proposed hexagonal acoustic metamaterial cell. (**a**) The whole model; (**b**) the gridded model; (**c**) the model of the hexagonal acoustic metamaterial cell; and (**d**) the gridded model of the hexagonal acoustic metamaterial cell.

**Figure 4 materials-16-05378-f004:**
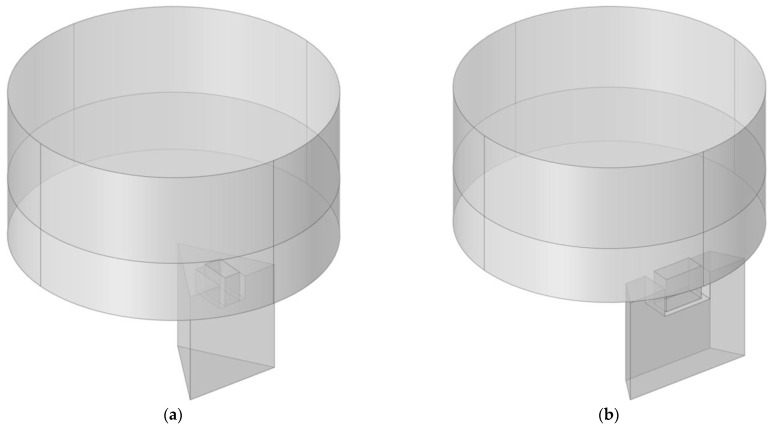
The acoustic finite element simulation model to study the sound absorption properties of a single cavity. (**a**) For the triangular cavity and (**b**) for the trapezium cavity.

**Figure 5 materials-16-05378-f005:**
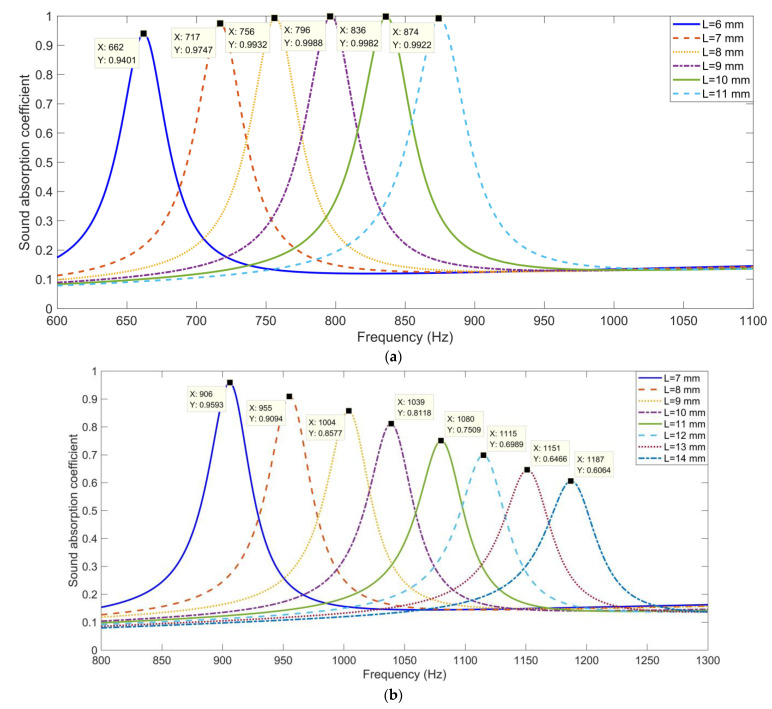
The sound absorption performances of single cavity with various parameters. (**a**) For the triangular cavity and (**b**) for the trapezium cavity.

**Figure 6 materials-16-05378-f006:**
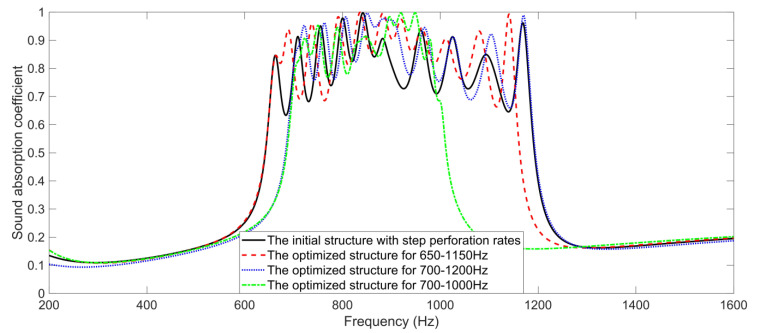
The theoretical sound absorption coefficients of the hexagonal acoustic metamaterial cell for certain specific scenarios.

**Figure 7 materials-16-05378-f007:**
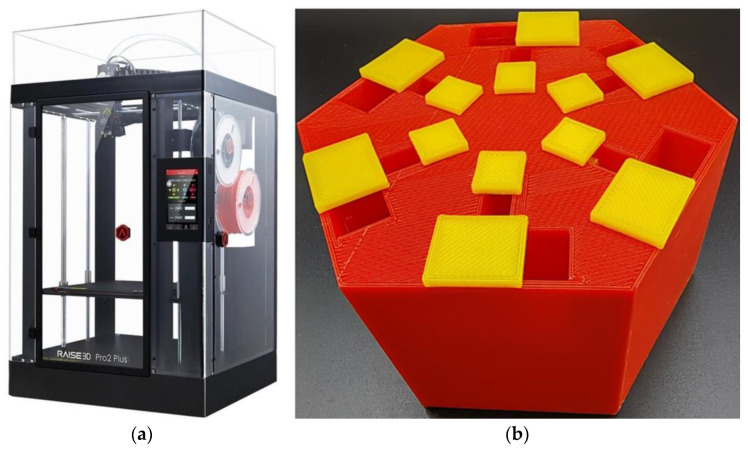
The preparation of the experimental sample for the hexagonal acoustic metamaterial cell. (**a**) The 3D printer and (**b**) the fabricated sample.

**Figure 8 materials-16-05378-f008:**
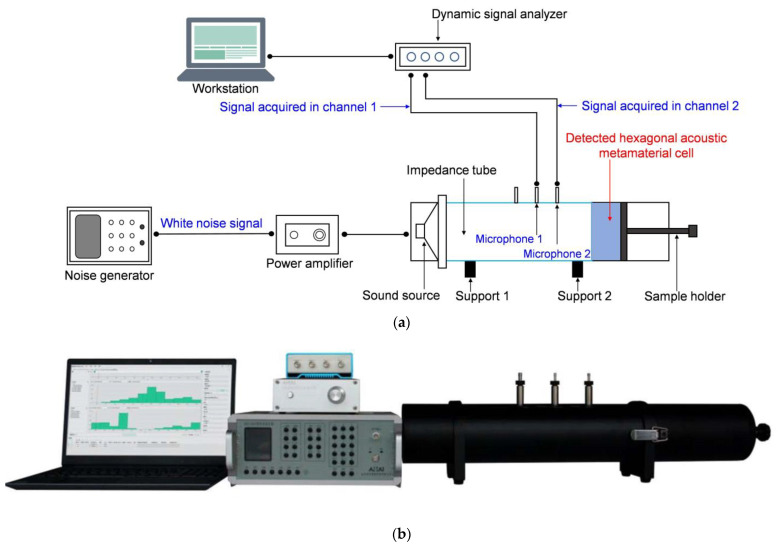
The detection of the actual sound absorption coefficients by the AWA6290T impedance tube detector; (**a**) the schematic diagram; and (**b**) the actual system.

**Figure 9 materials-16-05378-f009:**
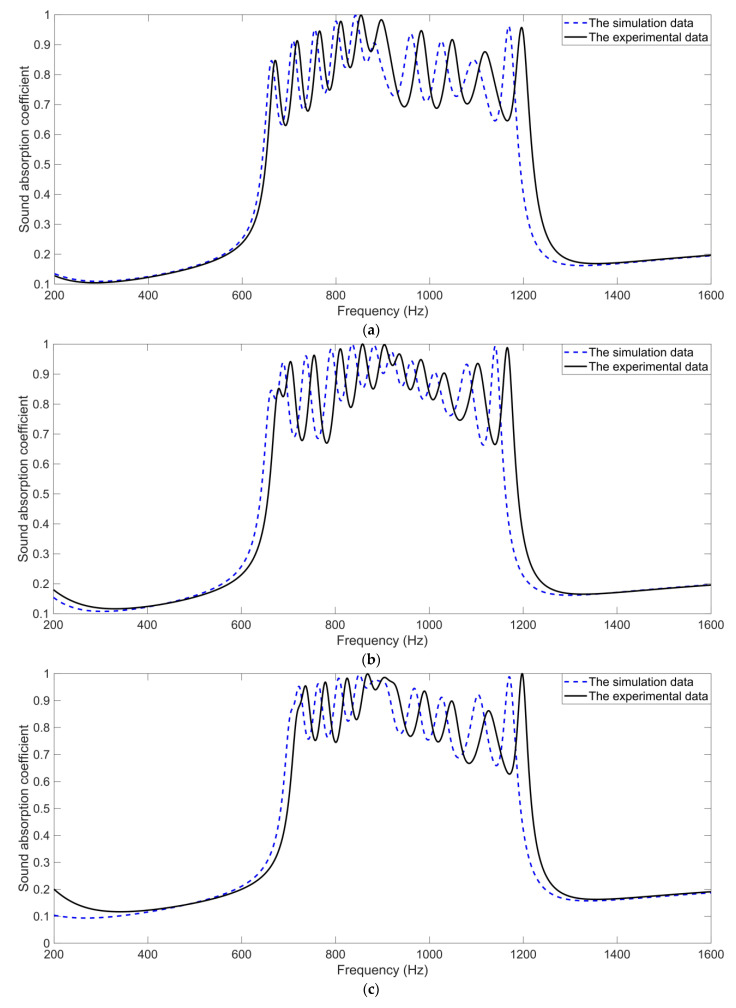
Comparative analysis of the sound absorption performance in the simulation with that of actual system. (**a**) The initial structure with step perforation rates; (**b**) the optimized structure for 650–1150 Hz; (**c**) the optimized structure for 700–1200 Hz; (**d**) the optimized structure for 700–1000 Hz.

**Figure 10 materials-16-05378-f010:**
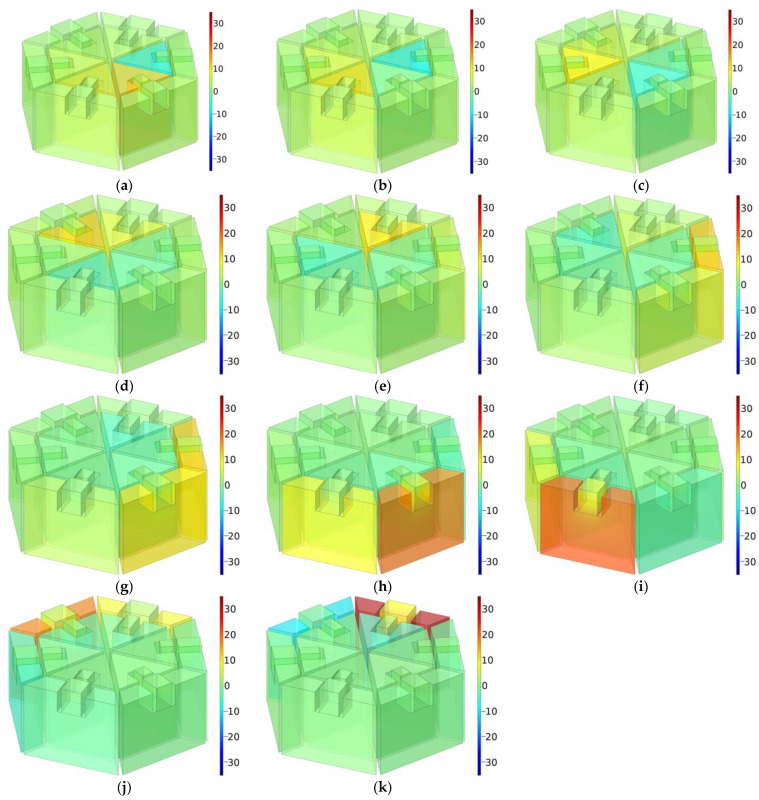
The distributions of sound pressure for the various resonance frequencies. (**a**) 664 Hz; (**b**) 689 Hz; (**c**) 737 Hz; (**d**) 791 Hz; (**e**) 836 Hz; (**f**) 882 Hz; (**g**) 920 Hz; (**h**) 962 Hz; (**i**) 1011 Hz; (**j**) 1080 Hz; and (**k**) 1140 Hz.

**Table 1 materials-16-05378-t001:** Summary of the resonance frequencies and corresponding sound absorption coefficients for the single cavity.

*L*	Triangular Cavity		Trapezium Cavity	
	Resonance Frequency	Peak Absorption Coefficient	Resonance Frequency	Peak Absorption Coefficient
6 mm	662 Hz	0.9401	–	–
7 mm	717 Hz	0.9747	906 Hz	0.9593
8 mm	756 Hz	0.9932	955 Hz	0.9094
9 mm	796 Hz	0.9988	1004 Hz	0.8577
10 mm	836 Hz	0.9982	1039 Hz	0.8118
11 mm	874 Hz	0.9922	1080 Hz	0.7509
12 mm	–	–	1115 Hz	0.6989
13 mm	–	–	1151 Hz	0.6466
14 mm	–	–	1187 Hz	0.6064

**Table 2 materials-16-05378-t002:** Summary of the selected geometric parameters for the target sound absorption performance.

*L* _n_		The Initial Structure with Step Perforation Rates	Target Sound Absorption Frequency Range		
			650–1150 Hz	700–1200 Hz	700–1000 Hz
Triangular cavity	*L* _1_	6.0 mm	6.0 mm	6.9 mm	6.9 mm
	*L* _2_	7.0 mm	6.6 mm	7.3 mm	7.3 mm
	*L* _3_	8.0 mm	7.6 mm	8.2 mm	7.9 mm
	*L* _4_	9.0 mm	8.8 mm	9.2 mm	8.8 mm
	*L* _5_	10.0 mm	9.8 mm	10.2 mm	9.7 mm
	*L* _6_	11.0 mm	11.0 mm	11.0 mm	10.0 mm
Trapezium cavity	*L* _7_	7.0 mm	7.5 mm	7.4 mm	7.0 mm
	*L* _8_	8.4 mm	8.4 mm	8.6 mm	7.4 mm
	*L* _9_	9.8 mm	9.5 mm	9.9 mm	8.0 mm
	*L* _10_	11.2 mm	10.8 mm	11.5 mm	8.7 mm
	*L* _11_	12.6 mm	12.0 mm	12.8 mm	9.1 mm
	*L* _12_	14.0 mm	13.2 mm	14.1 mm	9.4 mm

## Data Availability

The data that support the findings of this study are available from the corresponding author upon reasonable request.

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
