# Peer review of "Study on a Hexagonal Acoustic Metamaterial Cell of Multiple Parallel-Connection Resonators with Tunable Perforating Rate"

_materials, 2023, doi:10.3390/ma16155378_

Round 1

Reviewer 1 Report

Metamaterials are interesting applications in the field of applied acoustics.

The reference to health problems caused by noise and the increase in conflicts between people due to excessive noise is very interesting.

But in the introduction on metamaterials, a reference should be made to the origins of Veselago, Pendry. Furthermore, in the introduction I would make a reference to the recent studies on the metamaterials Alu, Ciaburro,  Cummer, Bevilacqua, Gupta, Trematerra.

Did you introduce the solution with a FEM model, did you use a commercial software?

In the measurements with impedance tube you built the 3D model: should you explain better how you performed the measurements with "AWA6290T impedance tube" what are the dimensions? Your sample as entered, impedance tube measurements can be affected by errors if you do not take into account the distances between the microphones, air gaps.

You have carried out the measurements on a sample of small dimensions and for normal incidence, what happens for a sample in a real case for diffuse incidence?

How does your numerical model adapt to full scale real?

Author Response

Response to reviewer’s comments

General Comment: Metamaterials are interesting applications in the field of applied acoustics.

Response:

Thank you very much for your kind review to our manuscript and constructive suggestions to our research. We have revised the manuscript carefully according to your and other reviewers’ meaningful comments. The corresponding responses to your comments one by one are as follows.

  1. The reference to health problems caused by noise and the increase in conflicts between people due to excessive noise is very interesting. But in the introduction on metamaterials, a reference should be made to the origins of Veselago, Pendry. Furthermore, in the introduction I would make a reference to the recent studies on the metamaterials Alu, Ciaburro, Cummer, Bevilacqua, Gupta, Trematerra.

Response:

Thank you very much for your valuable suggestion. A reference about the metamaterial with the variable geometric characteristics arranged in free space and excited in a waveguide radiator is added in revised manuscript, as shown in the new reference [17] in the revised manuscript. Meanwhile, according to your suggestion, a reference to the recent studies on the metamaterials is added in the new reference [18] in the revised manuscript. Furthermore, some relative literatures are added as well, which is shown in the new references [19] and [20] in the revised manuscript. Moreover, the sequence numbers of references are rearranged in the revised manuscript.

[17] Veselago, V.G.; Vinogradov, E.A.; Golovanov, V.I.; Zhukov, A.A.; Romanov, A.A.; Kapustyan, A.V.; Urlichich, Y.M.; Lav-rishchev, V.P. Waveguide propagation of microwave radiation in two-layer metamaterial. Tech. Phys. Lett. 2011, 37(3), 220-222.

[18] Iannace, G.; Ciaburro, G.; Trematerra, A. Metamaterials acoustic barrier. Appl. Acoust. 2021, 181, 108172.

[19] Hunt, J.; Driscoll, T.; Mrozack, A.; Lipworth, G.; Reynolds, M.; Brady, D.; Smith, D. Metamaterial apertures for computational imaging. Science 2013, 339, 310–313.

[20] Shitrit, N.; Yulevich, I.; Maguid, E.; Ozeri, D.; Veksler, D.; Kleiner, V.; Hasman, E. Spin-Optical Metamaterial Route to Spin-Controlled Photonics. Science 2013, 340, 724–726.

  1. Did you introduce the solution with a FEM model, did you use a commercial software?

Response:

Thank you very much for your meaningful question. The commercial COMSOL Multiphysics 5.5 software is utilized in this study, which can investigate the sound absorption performance of acoustic metamaterial and exhibit its sound absorption mechanism intuitively based on the acoustic finite element simulation. Meanwhile, the presentation of the used commercial COMSOL Multiphysics 5.5 software is added in the revised manuscript and highlighted in yellow.

  1. In the measurements with impedance tube you built the 3D model: should you explain better how you performed the measurements with "AWA6290T impedance tube" what are the dimensions? Your sample as entered, impedance tube measurements can be affected by errors if you do not take into account the distances between the microphones, air gaps.

Response:

Thank you very much for your useful question. Some details have been added in the description of sound absorption coefficient detecting process in the section 2.6, such as the distance between sample and the microphone 2, the distance between these 2 microphones, the sampling interval, etc. There were no air gaps in this detecting process, because the hexagonal acoustic metamaterial cell was installed at the end of the impedance tube and it was supported by the sample holder. The distance between the detected hexagonal acoustic metamaterial cell and the micro-phone 2 was kept as 100 mm, and that between the 2 microphones was selected as 70 mm, which realized the investigated frequency range of 200–1600 Hz according to the standard of the GB/T 18696.2–2002 (ISO 10534–2:1998) ‘Acoustics–Determination of sound absorption coefficient and impedance in impedance tubes–part 2: Transfer function method’. By this method, the actual sound absorption coefficients at the 1502 sampling frequency points in the frequency range of 200–1600 Hz could be obtained. Meanwhile, the added presentations are highlighted in yellow in the revised manuscript.

  1. You have carried out the measurements on a sample of small dimensions and for normal incidence, what happens for a sample in a real case for diffuse incidence?

Response:

Thank you very much for your meaningful question. Limited by this selected AWA6290T impedance tube detector, the sound absorption performance with the normal incidence was investigated in this research, and that with the diffuse incidence cannot be measured directly by this apparatus. The sound absorption performance with oblique incidence required the modified detector or the reverberation chamber method. For the classical transfer function tube detector, the measurement of sound absorption performance with oblique incidence required the incident tube, the sample, the reflected tube with the rear perfect absorbing material, and the intersection angle between the incident tube and reflected tube should be adjusted to be equal to the oblique incidence angle. Meanwhile, the normal line of detected sample should be consistent with the intersection angle between incident tube and reflected tube. The measurement in reverberation chamber required a sample with a very large size. In our previous study corresponding to the reference [21] “Yang, X.; Yang, F.; Shen, X.; Wang, E.; Zhang, X.; Shen, C.; Peng, W. Development of Adjustable Parallel Helmholtz Acoustic Metamaterial for Broad Low-Frequency Sound Absorption Band. Materials 2022, 15, 5938.”, we have studied the sound absorption performance of the adjustable parallel Helmholtz acoustic metamaterial with oblique incidence (the angle was 5–20° with the interval of 5°) by the acoustic finite element simulation, and the results were shown in the following figures. It could be found that the sound absorption performance was slightly improved along with increase of the incidence angle from 0° to 20°. which was consistent with the similar conclusions obtained in the literatures. In the literature “Wang, C.Q.; Huang, L.X.; Zhang, Y.M. Oblique incidence sound absorption of parallel arrangement of multiple microperforated panel absorbers in a periodic pattern. J. Sound Vib. 2014, 333, 6828–6842.”, Wang et al. had proved that equivalent acoustic impedance would decrease gradually with increase of the incidence angle, which changed the acoustic impedance matching conditions. For the oblique incidence with angle smaller than 45°, the change of the sound absorption performance was not obvious, no matter for the resonance frequencies or the peak sound absorption coefficient. Therefore, in a real case for the diffuse incidence within a reasonable range of the incidence angle (normally limited in 45°), the sound absorption performance of the sound absorbing materials or structures will not be affected observably, no matter for the resonance frequencies or the peak sound absorption coefficient.

Figure 1. The sound absorption performance with variable oblique incidence angle obtained in simulation.

  1. How does your numerical model adapt to full scale real?

Response:

Thank you very much for your helpful question. The hexagonal acoustic metamaterial cell is an extensive structure and it can be expanded in the plane direction. Thus, the sound absorption performance of each cell can represent that of the actual sample in the full scale real. Meanwhile, the consistency between the simulation data and the experimental data proved the accuracy of the acoustic finite element simulation model and the effectiveness of the joint optimization method in this research. Therefore, the developed hexagonal acoustic metamaterial cell had the advantages of adjustable sound absorption performance, excellent low frequency noise reduction characteristics, extensible outline structure, and efficient space utilization, which was conducive to promote its practical applications in the field of sound absorption for the varying noise spectrum.

Reviewer 2 Report

The presented manuscript seems to be interesting for readers of the Materials journal, it is written in a good manner and suits the requirements of the journal. It can be accepted for publication after minor corrections listed below.

- The "Abstract" section should contain the main achievements of research not general discussion. Re-organization of abstract is needed.

- State of the art to be improved further.

- What was the basis for choosing the parameters? Why are the experimental design methods not used in the selection of variables and their levels?

- In “Figure 2. The size of the hexagonal acoustic metamaterial cell”, the numbers are not legible.

- References 35 and 42; 36 and 41; 18 and 52 are duplicates. Please check and correct.

- All parameters used in formulas must be explained. It is recommended to attach all parameters and abbreviations used in a table at the end of the article.

- Abbreviation/ acronyms, should all be defined at their first occurrence in the manuscript,

- In the "Conclusion" section, the authors should present more quantitative data as the main results of the research study rather than just some qualitative data.

Minor editing of English language required

Author Response

Response to reviewer’s comments

General Comment: The presented manuscript seems to be interesting for readers of the Materials journal, it is written in a good manner and suits the requirements of the journal. It can be accepted for publication after minor corrections listed below.

Response:

Thank you very much for your kind review to our manuscript and constructive suggestions to our research. We have revised the manuscript carefully according to your and the other reviewers’ helpful comments. The responses to your comments are as follows.

  1. The "Abstract" section should contain the main achievements of research not general discussion. Re-organization of abstract is needed.

Response:

Thank you very much for your valuable suggestion. According to the template supplied by the journal, the abstract should contain the background, methods, results and conclusions of the whole research, and it should be limited in 200 words maximum, as shown in the following figure.

Figure 1. The template of the abstract supplied by the journal.

Therefore, the abstract in this study is organized as follows according to the template.

Background: The limited occupied space and various noise spectrum required an adjustable sound absorber with the smart structure and tunable sound absorption performance.

Methods: The hexagonal acoustic metamaterial cell of the multiple parallel–connection resonators with tunable perforating rate was proposed in this research, which consisted of 6 triangular cavities and 6 trapezium cavities, and the perforation rate of each cavity was adjustable by moving the sliding block along the slideway.

Results: The optimal geometric parameters were obtained by the joint optimization of the acoustic finite element simulation and cuckoo search algorithm, and the average sound absorption coefficients in the target frequency ranges of 650–1150 Hz, 700–1200 Hz and 700–1000 Hz were up to 0.8565, 0.8615 and 0.8807 respectively. The experimental sample was fabricated by the fused filament fabrication method, and its sound absorption coefficients were further detected by impedance tube detector. The consistency between simulation data and experimental data proved the accuracy of the acoustic finite element simulation model and the effectiveness of the joint optimization method.

Conclusions: The tunable sound absorption performance, outstanding low frequency noise reduction property, extensible outline structure and efficient space utilization were favorable to promote its practical applications in noise reduction.

Meanwhile, the whole abstract is polished to eliminate the spelling or grammar errors, and the corrections are highlighted in yellow in the revised manuscript.

  1. State of the art to be improved further.

Response:

Thank you very much for your meaningful comment. State of the art for the whole manuscript was further improved according to your and other reviewers’ comments. Through comparing with the other acoustic metamaterials in the introduction section, the advancement and advantage of this proposed hexagonal acoustic metamaterial cell of the multiple parallel–connection resonators with tunable perforating rate are exhibited. The corresponding added presentations are highlighted in yellow in the revised manuscript.

  1. What was the basis for choosing the parameters? Why are the experimental design methods not used in the selection of variables and their levels?

Response:

Thank you very much for your worthy question. There were 2 kinds of parameters for proposed hexagonal acoustic metamaterial cell. First was the selected parameter, such as the length of side of the hexagonal acoustic metamaterial cell, the width of each wall in the chamber, the width and thickness of the aperture for the 6 triangular cavities and those for the 6 trapezium cavities, etc. These parameters were selected according to the actual requirements of the noise reduction and the experimental measurement. Second was the adjustable parameter, which consisted of the lengths of rectangle apertures for the 12 cavities. In this study, the optimization of the adjustable parameter was conducted through the joint optimization of the acoustic finite element simulation and the cuckoo search algorithm instead of the experimental design methods in the selection of variables and their levels. The normal experimental design method, such as the orthogonal experimental method, is suitable for the condition that the influencing laws of parameters are not obvious. In this study, the influencing laws of parameters to the sound absorption coefficients are on a regular basis, so the initial values of parameters can be determined by the desired sound absorption properties, and their optimal parameters can be derived by the iterative optimization in the cuckoo search algorithm. Therefore, the joint optimization of the acoustic finite element simulation and the cuckoo search algorithm was applied to obtain the optimal geometric parameters of the length of the apertures in this research, which could promote the optimization efficiency and improve the optimization accuracy. These presentations are added about the comparisons of optimization efficiency and accuracy between the joint optimization method and the experimental design method, which are highlighted in yellow in the revised manuscript.

  1. In “Figure 2. The size of the hexagonal acoustic metamaterial cell”, the numbers are not legible.

Response:

Thank you very much for your valuable comment. Relative to the Figures 2a, 2b and 2d, the numbers in the Figures 2c and 2e are difficult to recognize. Thus, the Figures 2c and 2e are corrected to make them easy to identify, and the corrected figures are highlighted in yellow in the revised manuscript.

  1. References 35 and 42; 36 and 41; 18 and 52 are duplicates. Please check and correct.

Response:

Thank you very much for your kindly comment. Sorry for our mistakes. These repetitive references are replaced by the appropriate references. The initial references 41, 42 and 52 are replaced by the new references 45, 46 and 56 in the revised manuscript, as shown in the following literatures.

[45] Wei, W.; Ren, S.; Chronopoulos, D.; Meng, H. Optimization of connection architectures and mass distributions for metamaterials with multiple resonators. J. Appl. Phys. 2021, 129, 165101.

[46] Langfeldt, F.; Hoppen, H.; Gleine, W. Resonance frequencies and sound absorption of Helmholtz resonators with multiple necks. Appl. Acoust. 2019, 145, 314–319.

[56] Ding, C.; Hao, L.; Zhao, X. Two-dimensional acoustic metamaterial with negative modulus. J. Appl. Phys. 2010, 108, 074911.

  1. All parameters used in formulas must be explained. It is recommended to attach all parameters and abbreviations used in a table at the end of the article.

Response:

Thank you very much for your meaningful suggestion. A table is added as the appendix A in the revised manuscript, which described the meanings of each parameters used in this research.

  1. Abbreviation/ acronyms, should all be defined at their first occurrence in the manuscript.

Response:

Thank you very much for your valuable reminder. All the abbreviations and acronyms are defined at their first occurrence in the revised manuscript, and these revisions are highlighted in yellow.

  1. In the "Conclusion" section, the authors should present more quantitative data as the main results of the research study rather than just some qualitative data.

Response:

Thank you very much for your worthy suggestion. We have corrected the Conclusion section according to your comment, and some quantitative results are added to replace the qualitative presentations. These modifications are highlighted in yellow in the revised manuscript.

Round 2

Reviewer 2 Report

All concerns have been well addressed by the authors. Therefore, I recommend to publish the manuscript in its present form.

 Minor editing of English language required